# Review of the Tumor Microenvironment in Basal and Squamous Cell Carcinoma

**DOI:** 10.3390/cancers15092453

**Published:** 2023-04-25

**Authors:** Elizabeth Chiang, Haleigh Stafford, Jane Buell, Uma Ramesh, Moran Amit, Priyadharsini Nagarajan, Michael Migden, Dan Yaniv

**Affiliations:** 1School of Medicine, Baylor College of Medicine, Houston, TX 77030, USA; 2Head and Neck Surgery Department, The University of Texas MD Anderson Cancer Center, Houston, TX 77030, USA; dyaniv@mdanderson.org; 3Graduate School of Biomedical Sciences, The University of Texas, Houston, TX 77030, USA; 4Department of Pathology, The University of Texas MD Anderson Cancer Center, Houston, TX 77030, USA; 5Department of Dermatology, The University of Texas MD Anderson Cancer Center, Houston, TX 77030, USA

**Keywords:** tumor microenvironment, basal cell carcinoma, squamous cell carcinoma, immunotherapy

## Abstract

**Simple Summary:**

Non-melanoma skin cancers, including basal cell carcinoma and squamous cell carcinoma, are the most common types of cancer in the United States. In recent years, research into the tumor microenvironment of these cancers has shed light on the intricate network of cellular and acellular components surrounding tumor cells. This review aims to provide a deeper understanding of the complex interactions that occur within the tumor microenvironment and suggest potential therapies.

**Abstract:**

It is widely known that tumor cells of basal and squamous cell carcinoma interact with the cellular and acellular components of the tumor microenvironment to promote tumor growth and progression. While this environment differs for basal and squamous cell carcinoma, the cellular players within both create an immunosuppressed environment by downregulating effector CD4+ and CD8+ T cells and promoting the release of pro-oncogenic Th2 cytokines. Understanding the crosstalk that occurs within the tumor microenvironment has led to the development of immunotherapeutic agents, including vismodegib and cemiplimab to treat BCC and SCC, respectively. However, further investigation of the TME will provide the opportunity to discover novel treatment options.

## 1. Introduction

Skin cancer is the most common cancer diagnosed in the United States and represents a worldwide threat as its incidence steadily rises [1]. Risk factors for skin cancer development include both genetic predisposition and environmental factors, specifically, exposure to UV rays [2]. Basal cell carcinoma and squamous cell carcinoma are the most common skin cancer types [3]. 

The tumor microenvironment (TME) refers to the cellular environment of tumors or cancer stem cells [4,5]. In addition to tumor cells themselves, the TME includes nerve cells, epithelial cells, fibroblasts, plasmacytoid cells, dendritic cells, Langerhans cells, macrophages, and lymphocytes (Figure 1). This composition varies based on the type of cancer and a patient’s immune status [6]. Each of these cell types may act in either an anti- or pro-oncogenic manner, depending on their interactions with other components of the TME [7]. In turn, tumor cells stimulate cellular and molecular crosstalk in the TME to promote an immunosuppressive state, allow tumor progression, and even change the phenotype of cells in the TME [8,9]. 

While the TME has been studied across many types of cancer, this review aims to examine the role of the TME in non-melanoma skin cancer as well as potential therapeutic considerations. A better understanding of the intricate components of the TME in skin cancer will allow for the development of novel therapies to treat advanced skin cancers. 

## 2. Basal Cell Carcinoma

Basal cell carcinoma (BCC) is the most prevalent skin cancer, making up approximately 80% of all non-melanoma skin cancer cases [10]. Patients are typically over 60 years old at first presentation with BCC, and these lesions tend to present on body areas exposed to the sun. BCC pathogenesis is complex and has been traced to several risk-factors. Sun exposure is the usual culprit; however, recent research has connected increased physiologic exposure to hormone replacement therapy (HRT) and oral contraceptives (OCs) to aggressive BCC subtypes [11,12]. The molecular mechanism behind this connection is not well understood, but, conventionally, BCC pathogenesis has been traced back to aberrant sonic hedgehog (Shh) signaling [10]. Irreversible activation of Shh signaling creates high levels of oncogenic glioma-associated oncogene homolog (GLI) transcription factors, which initiate and promote BCC tumor growth [13]. Activation of Shh signaling in the TME has been associated with tumor growth and metastatic activity via its contribution to an immunosuppressed environment [10]. The BCC tumor develops the TME in a dense surrounding fibromyxoid stroma, protecting the tumor from the host’s immune system and promoting tumor angiogenesis and progression [14,15].

### 2.1. Cellular Components of the TME in BCC

Within the TME of BCC lesions, the most prevalent cellular players include cancer-associated fibroblasts, tumor-associated macrophages, CD4+ and CD8+ T cells, regulatory T cells, and dendritic cells, as well as their differentiated subset Langerhans cells [16,17,18].

Cancer-associated fibroblasts (CAFs) are an activated and differentiated form of normal tissue-resident fibroblasts induced by chronic UV exposure and signaling from the tumor [17,19]. Abundant in the BCC TME, CAFs secrete various chemokines, cytokines, and extracellular matrix proteins that downregulate the host’s anti-tumor response [17,20,21]. For example, CAFs secrete CXCL12 and CCL22 to prevent invasion of CD4+ and CD8+ T cells and instead recruit regulatory T cells (Tregs) to the TME [17]. To accomplish this, CAFs can develop a multitude of different phenotypes to form a heterogeneous population, demonstrating plasticity in the TME [22].

Macrophages exist in either an anti-tumorigenic M1 form or a pro-tumorigenic M2 form. Generally, M1 macrophages protect the host from infections and tumor progression, while M2 macrophages promote tumor growth [23]. Tumor-associated macrophages (TAMs) are a subtype of M2 macrophages that promote tumor growth, invasion, and metastasis and activate tumor-promoting genes in BCC lesions [24]. Indeed, when evaluating the BCC microenvironment, Beksac et al. discovered a higher volume of M2 macrophages than other macrophages, suggesting an M2-dominant, pro-oncogenic TME [16].

It is widely known that BCC cells influence their surroundings to promote tumorigenesis; however, CAFs and TAMs can also directly induce changes in the tumor cells. Lacina et al. demonstrated that CAFs taken from the TME of BCC can influence normal keratinocytes toward malignant growth characteristics and phenotypes when they are cultured together in situ. This suggests that stromal cells in the TME have a regulatory role in BCC progression [20]. TAMs also can influence BCC development by directly activating tumor-promoting genes in BCC cells, leading to the induction of tumor invasion and angiogenesis [24]. Analysis of whole-genome RNA-seq data of BCC tumor microenvironments revealed that samples of more advanced BCCs had higher concentrations of TAMs, increasing tumoral inflammation [14].

The TME consists of many tumor-infiltrating lymphocytes, particularly Tregs. Tregs are a subtype of CD4+ helper T cells that control autoimmunity by suppressing conventional T cells, creating an immunosuppressive environment that promotes tumor progression [18,25]. For instance, in comparing the TME of a BCC lesion to the microenvironment of normal non-UV-exposed buttock skin, Omland et al. discovered a high concentration of Tregs in the peritumoral skin, while no Treg expression was found in the normal skin sample. Moreover, Tregs comprise 8–20% of the CD4+ T cell population in normal adult skin, but they account for approximately 45% of CD4+ cells in the BCC TME, demonstrating a two-fold increase that mediates the immunosuppressed environment to favor skin cancer progression [25]. However, by using RNA-Sequencing data to explore the BCC TME, Lefrançois et al. did not observe more Tregs in BCC when compared to normal skin [4]. This difference in results reflects the ongoing controversy surrounding the influence of Tregs on growth patterns in BCC [15]. On the other hand, CD8+ T cells are known to exert an anti-tumor response; however, the TME has a lower volume of CD8+ cells than normal skin does, further promoting immunosuppression that allows for tumor growth [16,26]. This volume might change in organ transplant patients receiving immunosuppressive drugs. Using state-of-the-art nonlinear image analysis techniques such as fractal dimension and sample entropy of internuclear distances and comparing tissue microarchitecture and inflammatory infiltrates of BCC in kidney transplant patients to healthy controls, Capasso et al. found that there is a 70% increase in the density of inflammatory cells in the kidney transplant patients’ samples. Since tumors still develop in this setting, this does not speak to the activity of these inflammatory cells [6].

Dendritic cells (DCs) and their epidermal subset Langerhans cells serve as antigen-presenting cells, beneficial for tumor eradication in the skin. These cells can recognize, process, and present antigens to T cells or stimulate T cell proliferation and activity [23,27]. Similar to the controversy surrounding the prevalence of Tregs in the BCC TME, there are conflicting results regarding the composition of DCs. Some studies indicate that the BCC microenvironment has high numbers of immature DCs, which are less effective at stimulating an anti-tumor immune response [18]. However, a recent study performed by Lefrançois et al. did not show more immature dendritic cell infiltration in BCCs compared to normal skin [14]. Regardless, CD1a-expressing DCs, cells typically responsible for polarizing naïve CD4+ T cells into an anti-tumor Th1 phenotype, were scarce in BCC peritumoral infiltrates [16,28].

Lastly, the lipid composition of the TME seems to affect the tumor’s aggressiveness. Dimoska Nilsson et al. performed maximum autocorrelation factor (MAF) analysis on data collected from a small area of aggressive BCC made up of more and less aggressive BCC tumor islands and inflammatory cells to target the chemical profile of the microenvironment of BCCs, where the signals were not dominated by lipid accumulations in glands or follicles. Increased knowledge about the specific lipid profiles in different cancers and in their microenvironment could lead to the development of targeted cancer therapies [29].

### 2.2. Signaling Pathways in the TME of BCC

In the BCC microenvironment, cytokines dominate pro-tumorigenic signaling between TME cellular components and the tumor. It has been well established that the BCC microenvironment predominantly consists of Th2 cytokines [14,18,24,26,30]. By studying BCC lesions with and without the topical immune modifier imiquimod, Tjiu et al. demonstrated that an environment abundant in Th2 cytokines, such as interleukin (IL)-4, IL-5, IL-10, and IL-13, facilitated BCC progression. In contrast, an environment with an imiquimod-induced Th1 response led to BCC regression, further supporting the pro-oncogenic power of a Th2-dominated environment [24]. Th2 cytokines induce an immunosuppressive environment surrounding the tumor site, contributing to BCC pathogenesis [31].

A myriad of cytokines circulate in the TME, promoting both anti-oncogenic and pro-oncogenic states; however, the most important factors surrounding BCC lesions include the cytokines interferon gamma (IFN-γ), TGF-β, IL-6, IL-10, and CCL22 (Figure 2) [18,26]. CD4+ and CD8+ T cells infiltrating the BCC microenvironment produce IFN-γ, suggesting a host antitumor response; however, these anti-tumorigenic T cells are downregulated by signaling from CAFs and Tregs [18,30]. TGF-β contributes to the induction of Tregs into the TME and the myofibroblastic differentiation of fibroblasts, both of which contribute to an immunosuppressive TME, thus inhibiting the elimination of tumor cells [15]. IL-6, is a pro-oncogenic cytokine that enhances anti-apoptotic activity and promotes angiogenesis in BCC. Elamin et al. discovered that IL-6 concentrations were significantly higher in the BCC TME than in that of squamous cell tumors [26]. Finally, IL-10 and CCL22, chemokines responsible for Treg development and chemotaxis, were also increased in the BCC TME [18,31].

The upregulation of these cytokines can be due to their roles in promoting immunosuppression, the primary function of the TME. Since CD4+ and CD8+ lymphocytes have profound anti-oncogenic effects, the TME cellular components work to prevent these T cells from activating and infiltrating the environment. CAFs engage in this activity directly by immobilizing T cells by secreting CXCL12 and indirectly by recruiting Tregs using CCL22. Tregs serve many vital functions in maintaining an immunosuppressed environment. Most importantly, they modulate the response of anti-tumorigenic CD4+ and CD8+ T cells [17]. They also attenuate the function of antigen-presenting DCs to promote immune system evasion [18]. Lastly, they secrete IL-10, recruiting even more Tregs to the BCC microenvironment [25]. Thus, the crosstalk within the TME and between TME components and tumor cells promotes BCC’s progression and immune evasion within the host.

Ultimately, the BCC tumor cells greatly impact the TME. Through Shh/GLI signaling, BCC tumor cells induce immunosuppressive mechanisms, including the recruitment of Tregs, producing IL-10 [13]. Moreover, the membrane-bound human leukocyte antigen (HLA) g has been detected on non-aggressive primary BCC tumor cells. HLA g induces immune inhibition by binding to immune cells to induce apoptosis of CD8+ T cells and NK cells or inhibit one or more processes, including the proliferation of CD4+ T cells, antigen presentation by antigen-presenting cells (APCs), and maturation of DCs [32]. Thus, the BCC cells directly influence the TME, which, in turn, provides the ideal conditions for tumor growth.

## 3. Squamous Cell Carcinoma

Cutaneous squamous cell carcinoma (SCC) is the second most common type of non-melanoma skin cancer, accounting for approximately 20% of skin cancers [33]. Risk factors for developing SCC include advanced age, extensive sun exposure, and immunosuppression [33]. Typically, most SCC lesions can be successfully treated with excision; however, some metastasize, potentially becoming life-threatening. Thus, about 75% of all deaths due to skin cancer are caused by SCC [33]. The development of SCC is gradual, stemming from an uncontrolled proliferation of epidermal keratinocytes and progression through dysplasia and actinic keratosis as the tumor cells accumulate genetic mutations in genes such as *TP53*, *CDKN21*, *NOTCH*, and *RAS* [33,34]. During tumorigenesis, the cancer cells enable growth, angiogenesis, and metastasis through autocrine and paracrine signaling [35]. Recent research has shown that the loss of p53, which commonly occurs in SCC tumors, allows for increased expression of type 2 Deiodinase (D2), an enzyme that activates the thyroid hormone (TH) [36,37]. In turn, TH induces VEGF-A transcription in SCC tumor cells, fostering tumor angiogenesis, nutrient delivery, and cancer progression [38]. To further understand the progression of SCC, it is essential to review the components of its TME and their interactions.

### 3.1. Cellular Components of the TME in SCC

Similar to the TME of BCC, SCC has many of the same cellular components, including CAFs, TAMs, tumor-infiltrating lymphocytes, and DCs. However, the SCC TME also has a significant invasion of tumor-associated neutrophils.

In SCC, CAFs secrete several factors that can stimulate cancer cell proliferation, mainly by promoting inflammation. Erez et al. compared CAFs to normal human skin fibroblasts and discovered that CAFs derived from human SCC tumors expressed genes promoting an inflammatory signature, while regular fibroblasts did not [39]. Further studies have also demonstrated CAFs’ ability to induce and sustain inflammation, enhance SCC’s invasive properties, and promote angiogenesis [40].

Similar to in BCC, M2 macrophages predominate in SCC tumors, promoting an oncogenic environment. These macrophages have a lower antigen-processing capacity than M1 macrophages, which makes them less effective in stimulating an anti-tumorigenic immune response in SCC [41]. In addition, M2 macrophages promote tumor growth in the SCC TME by stimulating angiogenesis and tissue remodeling [42].

Again, tumor-infiltrating lymphocytes play critical roles in developing pro- and anti-oncogenic forces within the TME. Tregs promote an immunosuppressive environment around the SCC lesion by directly suppressing effector T cells and indirectly suppressing T cell activity through downregulating antigen-presenting cells, with high numbers of Tregs associated with less favorable outcomes in the treatment of SCC [43,44]. Research in other types of cancer, including BCC, shows that CD4+ T cells play an anti-tumorigenic role. Surprisingly, in cutaneous SCC, CD4+ T cells enhance immunosuppression, tumor development, and tumor growth [45]. The TME surrounding SCC lesions is infiltrated with a high proportion of CD4+ T cells, which would be associated with spontaneous regression in other cancers such as primary melanoma and BCC [46,47]. Therefore, it is likely that the CD4+ T cells surrounding SCC tumors have different, pro-oncogenic properties [48]. Using a mouse model, de Oliveira et al. discovered that CD4+ T cells in the SCC TME inhibit the activation and migration of natural killer cells [49]. Typically, natural killer cells are part of the innate immune system and keep tumor progression in check; therefore, inhibiting these cells increases susceptibility to carcinogenesis [50]. On the other hand, CD8+ T cells demonstrate an antitumor immune response through the production of IFN-γ [45,49]. However, Freeman et al. demonstrated that the percentage of CD8+ T cells within invasive SCC TME was lower than that of SCC in situ, indicating a depletion of CD8+ cells as SCC progresses [48].

Langerhans cells, an epidermal subtype of DCs, are the first antigen-presenting cells to encounter SCC tumor antigens and present them to T cells in local lymph nodes [27,41]. However, elevated concentrations of TGF-β in the SCC microenvironment have been linked to lower amounts of DC infiltration, leading to fewer DCs able to enter the lymphatic system and activate tumor-infiltrating T cells [27].

Similar to TAMs, tumor-associated neutrophils (TANs) have both pro-tumor and anti-tumor functions, with an N1 and N2 state. N1 TANs kill tumor cells or inhibit their growth while N2 TANs promote cancer cell invasion, metastasis, angiogenesis, and immunosuppression [51]. Polarization into either the N1 or N2 phenotype is controlled by the expression of TGF-β, with higher levels promoting the pro-oncogenic N2 form [51,52]. Notably, Seddon et al. proved that patients with elevated blood neutrophil counts were more likely to develop cutaneous SCC lesions that exhibited high-risk tumor features, demonstrating an association between increased neutrophil levels and poor SCC outcomes [53]. This phenomenon likely occurred because N2 TANs in SCC promote immunosuppression by limiting CD8+ T cell responses via arginine depletion in the TME. Arginine is a metabolite crucial for T cell effector function; thus, by depleting the arginine supply, TANs downregulate the activity of CD8+ T cells. In addition, N2 TANs upregulate PD-L1; thus, they can bind to CD8+ T cell PD-1 in the TME, suppressing T cell responses through the PD-L1/PD-1 immune checkpoint interaction [51].

### 3.2. Signaling Pathways in the TME of SCC

Again, cytokines exert both pro- and anti-oncogenic effects to influence cells of the TME (Figure 3). The cells within the SCC TME respond to a dynamic mix of Th1 and Th2 signals. In the SCC TME, Pettersen et al. found a strong IFN-γ genomic signature and increased IFN-γ activation of infiltrating TAMs, indicating a Th1-type immune environment [42]. However, CD4+ T cells in the TME secrete IL-4, IL-10, and IL-17, which are Th2-type cytokines [42,45]. Given that a Th1-dominant environment inhibits tumor progression, the presence of both Th1 and Th2 signaling demonstrates the complex interplay of opposing cytokines that ultimately leads to a Th2-dominant TME.

As with other cancers, many key cellular players in the SCC TME secrete cytokines that contribute to the pro-oncogenic Th2 state. CAFs produce matrix metalloproteinases (MMPs) and cytokines, including VEGF and IL-6, which enhance tumor growth and progression in SCC [40]. TAMs express MMP-9 and MMP-11, which contribute to the poor prognosis of tumor-bearing hosts by increasing lymphatic vessel density, potentially enabling metastasis [41,42,43]. In addition, Tregs may produce IL-10 and TGF-β, promoting immunosuppression in the SCC TME [49].

Another important contributor to signaling in the TME of SCC is specialized membranous extracellular vesicles (EVs), which can transport microRNAs (miRNAs) that alter pro-oncogenic pathways. Flemming et al. discovered that Desmoglein 2 (Dsg2), a protein component of desmosomes, was highly expressed in non-melanoma skin cancers and regulated EV biogenesis and secretion from virtually every cell type, including keratinocytes, fibroblasts, and lymphocytes. In SCC, Dsg2 promotes tumor development and progression by increasing the production of ICAM-1 and IL-8, which inhibit leukocyte binding and promote inflammation and, ultimately, tumor progression [35].

Finally, SCC tumor cells overexpress TGF-β, an immunosuppressive cytokine that significantly blocks immune responses and facilitates tumor progression [52]. De Oliveira et al. demonstrated that the transfer of Tregs into the SCC TME increased the tumor cells’ production of TGF-β, which inhibited the proliferation of effector T cells, promoting an immunosuppressed environment [49]. TGF-β also modulates the activity of DCs, affecting both DC invasion and escape from the TME. Weber et al. showed that transfection of TGF-β into SCC cells decreased the number of infiltrating DCs by approximately 25%. In addition, transfection led to reduced infiltration by CD4+ and CD8+ T cells, likely due to TGF-β–mediated prevention of DC exit and presentation of antigens to lymph node-residing T cells [27]. TGF-β also plays a vital role in defining the TAN phenotype, skewing differentiation toward the N2 pro-oncogenic phenotype [52].

## 4. Therapeutic Considerations for Skin Cancer

For both BCC and SCC, surgical excision is the preferred treatment [54,55]. However, for advanced or metastatic non-resectable cancers, management has transitioned to an immunotherapeutic approach [56]. The treatment of advanced BCC focuses on Shh inhibition, while therapy for SCC interferes with the PD-1/PD-L1 checkpoint [57,58].

Early treatment of BCC with surgical resection is curative in most cases, but, for some patients, locally advanced or metastatic tumors can be life-threatening or, at the very least, can negatively impact a patient’s quality of life [54]. Therefore, clinicians have turned to Shh pathway inhibitors such as vismodegib or sonidegib as an alternative treatment for more severe cases [54,57]. Administration of these Shh inhibitors triggers increases in tumor-infiltrating CD8+ T cells and tumor-cell major histocompatibility complex I (MHC class I), stimulating an anti-tumoral response within the TME [57]. The ERIVANCE trial, studying the efficacy of vismodegib in advanced BCC, showed overall response rates of 48.5% in metastatic cohorts and 60.3% in locally advanced patients [59]. However, that still leaves many cases of BCC that are unresponsive to this treatment, despite its seemingly beneficial immunologic effects [57]. Recent studies have shown that vismodegib-resistant tumor cells transform their cell identity toward a mesenchymal stem cell-like profile, becoming resistant to Shh pathway inhibitors but providing a potential target for therapy [14].

For patients with SCC, surgical excision is the treatment modality of choice; however, for patients with increased risks of local recurrence, perineural spread, and metastasis, immunotherapy with cemiplimab, pembrolizumab, or nivolumab have been considered [55,58]. These drugs are PD-1 inhibitors, which inhibit the PD-1/PD-L1 immune checkpoint that typically decreases T cell functionality in the TME, suppresses the immune system, and accelerates cancer cell proliferation [58,60]. Therefore, treatment with these immunotherapy agents increases the volume of cytotoxic CD8+ T cells in the TME, allowing for tumor cell destruction [58]. In advanced-stage patients who responded to neoadjuvant cemiplimab, the drug had an overwhelming response, with many patients achieving complete pathologic responses [61]. The most suitable combination of this drug with other treatment modalities such as surgery and radiation, the clinical meaning of a pathologic complete response after immunotherapy, and the need for further treatment are still being explored. However, similar to the treatment of BCC with vismodegib, SCC therapy with cemiplimab achieved an objective response rate of approximately 50% of patients [62]. Therefore, further research must be conducted to determine more effective treatment methods for advanced SCC.

A better understanding of the TME provides many potential sites for intervention. For example, finding a way to polarize TAMs toward their M1 anti-tumorigenic phenotypes through immune modulation mechanisms may be a nonsurgical method to treat advanced BCC in the future [24]. Likewise, TANs in SCC could be polarized more toward the N1 phenotype through the inhibition of TGF-β in the TME [52]. As new research further explores the intricacies of the TME of BCC and SCC, there will be more opportunities to develop targeted therapies to treat advanced cancers.

Research into TME therapeutic targets is ongoing. Lu et al. discovered that the overexpression of the brain-derived neurotrophic factor (BDNF) and its p75 pan-neurotrophin receptor could induce BCC tumor cell death by stimulating M1 macrophages and T cell recruitment on a mouse model, providing a potential therapeutic intervention for Shh inhibitor-resistant tumors [63]. Moreover, by studying cancers with similar TME profiles to that of BCC, researchers have begun investigating the use of therapies for other cancers, such as kidney chromophobe cancers or myxofibrosarcoma, in treating advanced BCC. For example, Zhang et al. suggested targeting the RB pathway, as one would treat myxofibrosarcoma to modulate the TME contained within the BCC fibromyxoid stroma, rendering it less suitable for tumor growth [64]. We conducted a search on clinicaltrials.gov using the search terms “Skin Cancer, Non-Melanomta”, “SCC”, and “BCC” to identify clinical trials targeting components of the TME. Some of these innovative trials include a few institutions who are attempting to use genetically modified viruses to enhance host immune responses against BCC and cutaneous SCC lesions. Researchers are studying the type I herpes simplex virus genetically modified to preferentially replicate in tumor cells talimogene laherparepvec (T-VEC) because it can be injected directly into tumors to enhance antigen presentation by DCs, promoting an anti-oncogenic immune response [ID: NCT03458117, NCT02978625, NCT04163952] [65]. Other institutions are researching similar biologics including Imvamune, a modified smallpox vaccine, and IFx-Hu2.0, an immunomodulatory agent that triggers innate and adaptive immune responses in non-melanoma skin cancer tumors [ID: NCT04410874, NCT04160065]. Specifically for treating cutaneous SCC, a non-coding RNA called CV8102 has been developed to mimic a viral infection of the tumor, which serves to recruit and activate APCs and, subsequently, T cells to kill tumor cells at the site of injection [ID: NCT03291002]. Thus, a deeper understanding of the cellular players and interactions within the TME has allowed for the development of novel immunotherapies.

## 5. Conclusions

Exploration of the TME has provided insight into the crosstalk between cancer cells and the various components of the TME. While the TMEs of BCC and SCC vary slightly, the cellular and acellular players of both promote a pro-oncogenic environment through the downregulation of tumor-infiltrating lymphocytes. This creates an immunosuppressed environment ideal for tumor growth and metastasis; however, knowledge of the components of the TME that contribute to this pro-tumorigenic state has allowed for the implementation of novel therapeutics, including vismodegib and cemiplimab to treat BCC and SCC, respectively. While these immunotherapeutic agents have demonstrated efficacy in treating advanced non-melanoma skin cancers, further study of the TME will shed light on potential therapeutic targets representing future directions for treating cancer. In fact, several researchers have leveraged their understanding of the TME to develop immunotherapy agents, such as T-VEC and other viral infection imitators, designed to recruit anti-oncogenic immune cells, including DCs and cytotoxic T cells. Upregulation of these inflammatory cells in the TME will promote tumor cell destruction by inducing the host’s own defense system. In this manner, therapies developed around knowledge of the TME in BCC and SCC may provide an opportunity to cure advanced cancers resistant to existing treatments.

## Figures and Tables

**Figure 1 cancers-15-02453-f001:**
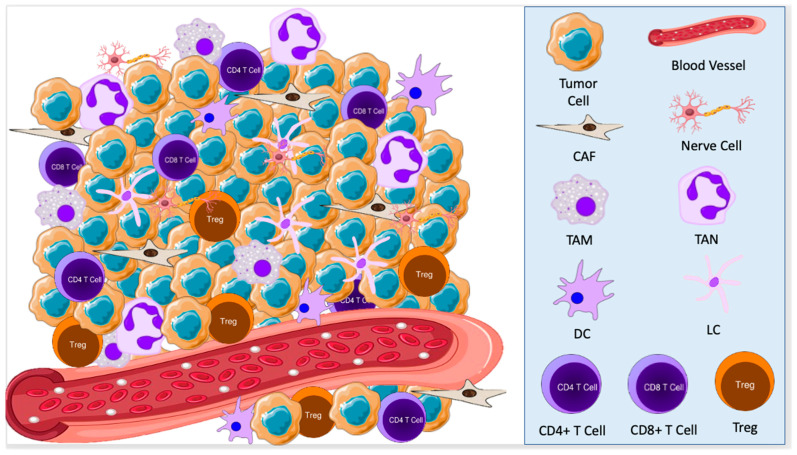
The tumor microenvironment consists of tumor cells, nerve cells, cancer-associated fibroblasts (CAFs), tumor-associated macrophages (TAMs), tumor-associated neutrophils (TANs), dendritic cells (DCs), Langerhans cells (LCs), and lymphocytes. These cells interact with each other to promote a pro-oncogenic state, allowing for tumor growth and development.

**Figure 2 cancers-15-02453-f002:**
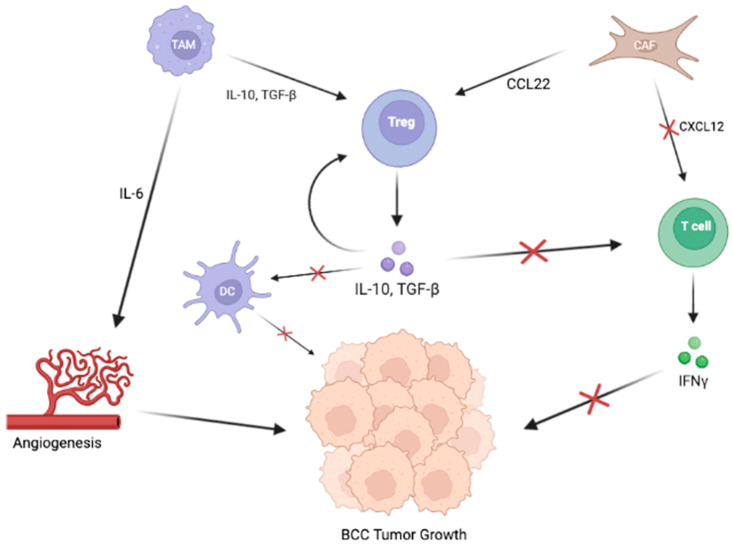
The cellular elements of the TME interact heavily with each other and BCC tumor cells to, ultimately, promote tumor growth by downregulating immune responses by dendritic cells and T cells. TAMs and CAFs upregulate Tregs through the secretion of IL-10, TGF-β, and CCL22, respectively. Tregs secrete IL-10 and TGF-β, which inhibit DCs and T cells from destroying BCC tumor cells. TAMs also secrete IL-6, which stimulates angiogenesis, supporting tumor growth and development.

**Figure 3 cancers-15-02453-f003:**
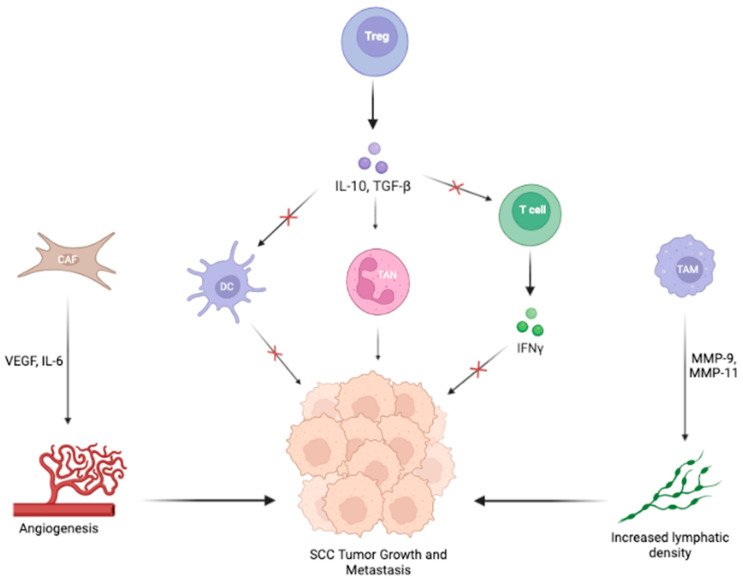
The cellular elements of the TME interact with each other and SCC tumor cells to promote growth and metastasis. Tregs secrete IL-10 and TGF-β, which downregulates DC and T cell activities, promoting an immunosuppressive environment for SCC development. In addition, these cytokines promote TAN activity, subsequently supporting tumor growth. CAFs produce VEGF and IL-6, which promote angiogenesis and tumor metastasis. TAMs secrete MMP-9 and MMP-11, increasing lymphatic vessel density and enabling metastasis.

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
