# Peer review of "(untitled)"

_cancers, 2023, doi:10.3390/cancers15092453_

Round 1

Reviewer 1 Report

This is an excellent review on the tumor microenvironment (TME) of BCC and cutaneous SCC. The review is logical and well organized, with useful pictures. In particular the SCC part has a thorough literature review.

There are several key recent references missing from the review according to a simple PubMed search, mostly in the BCC part (see below). Thank you for your attention to these newer references on the BCC TME.

  • 10.1080/20013078.2020.1790159

  • 10.1007/s12079-020-00563-6

10.1159/000507581

10.1002/1878-0261.12758

10.1116/6.0000340

10.1155/2021/6652846

10.1007/s00403-021-02218-x

10.1111/cup.14381

10.3390/cancers15010305

Author Response

Thank you very much for these insightful comments and suggestions. We have revised our paper to include the suggested articles as listed below:

10.1080/20013078.2020.1790159 – We incorporated this paper's signaling ideas briefly into the SCC introduction in lines 197-199. We mainly referenced this source in section 3.2, Signaling Pathways in the TME of SCC. Here, we added a paragraph on specialized membranous extracellular vesicles (EVs) and the role of Desmoglein 2 in regulating the synthesis of EVs to promote tumor progression through microRNAs (lines 274-282).

10.1007/s12079-020-00563-6 – This paper looked closely at the TME using RNA sequencing of the fibromyxoid stroma, which we incorporated into the introduction paragraph for BCC lines 63-66. We added cellular findings from this paper to several paragraphs in section 2.1 Cellular Components of the TME in BCC, including the higher concentration of TAMs in advanced BCCs (lines 96-98), no increase in Tregs (lines 108-110), and no increase in DCs (lines 128-130). This paper also provided insight into vismodegib-resistant BCC tumor cells, which was added to the section on therapeutic considerations (lines 310-313).

10.1159/000507581 – The unique insight of BCC in the setting of kidney transplant patients was added to section 2.1, Cellular Components of the TME in BCC. We used information from this paper to expand upon lymphocytes and inflammatory cells in the TME to give a richer picture of the environment in different patient populations (lines 114-121).

10.1002/1878-0261.12758 – We used this paper to enhance the introductory paragraph on BCC to provide background information on the pathogenesis of BCC tumor growth (lines 59-61). We also incorporated the idea of Shh/GLI signaling into section 2.2, Signaling Pathways in the TME of BCC, to explain one mechanism by which BCC tumor cells impact the TME through signaling (lines 178-180).

10.1116/6.0000340 – This paper examined the lipid composition of the TME and its impact on a tumor’s aggressiveness. We added a paragraph to incorporate the paper’s ideas into the section on cellular components of the TME of BCC, which helped convey the complexity of the TME since cancers with varying degrees of aggressiveness have different lipid profiles (lines 133-140).

10.1155/2021/6652846 – We used the information from this paper to expand section 4, Therapeutic Considerations for Skin Cancer (lines 337-341). The discovery of the overexpression of BDNF and its p75 pan-neurotrophin receptor creates an opportunity to develop a new therapy utilizing components of the TME, like M1 macrophages, to combat Shh inhibitor-resistant BCCs.

10.1007/s00403-021-02218-x – By investigating HLA-G in the context of BCC, this paper adds another mechanism for how BCC tumor cells impact the TME. We included that HLA-G induces immune inhibition by upregulating cellular components of the TME, such as CD8+ T cells and NK cells, and downregulating the activity of anti-oncogenic components like CD4+ T cells and DCs (lines 180-186).

10.1111/cup.14381 – We incorporated this paper into section 2.2, Signaling Pathways in the TME of BCC, to add more information about the effects of TGF-b  on the induction of Tregs, ultimately contributing to an immunosuppressed environment. This source helped clarify the intricate signaling pathways that occur (lines 158-161).

10.3390/cancers15010305 – This paper helped expand the section on therapeutic considerations for BCC. It suggested using treatments for cancers with similar TMEs to that of BCC to target BCC tumor cells resistant to more conventional treatment methods. We added this idea to lines 341-347.

Thank you again for this great feedback – we feel your suggestions helped enhance the paper in a substantial manner.

Reviewer 2 Report

The manuscript entitled “Systematic Review of the Tumor Microenvironment in Basal and Squamous Cell Carcinoma” by Elizabeth Chiang et al., was presented as a Literature Review in which the authors purposed to disset the role of the TME in two different type of skin cancers, the BCC and SCC. From the title of the manuscript the aim of the review is to dissect the crosstalk that occurs within the tumor microenvironment in order to develop immunotherapeutic strategies counteracting the tumor progression. However, the premise is not satisfied by the body of the review which, although presents the actual knowledge from published studies in a comprehensive and interesting way, seems to lack a real insight into the mechanisms involved in the various aspects of the role of the TME with mention of substitutive therapy/treatment options. The reviewer suggests to extend the section regarding the BCC and SCC pathogenesis, focusing on the latest knowledge about the genetic and hormonal signaling pathways involved in BCC and SCC tumor formation and progression. Moreover, the manuacript lacks a more detailed conclusions section. It would be necessary improve the conclusion, adding a short discussion section and focusing on the “Therapeutic Considerations for Skin Cancer”. Finally, the review needs to be condensed. A schematic/cartoon summarizing the signaling pathways in the TME of BCC and SCC is preferred. Therefore, the authors consider the possibility to add a graphical representation for both the sections 2.2 Signaling Pathways in the TME of BCC and 3.2 Signaling Pathways in the TME of SCC. 

Author Response

Thank you very much for these insightful comments and suggestions. We have revised our paper to address your feedback, as stated below.

Regarding the need for insight into the mechanisms involved in the various aspects of the role of the TME, we agree with your feedback and have tried to ameliorate the disconnect between the title and the body of the review. As suggested, we expanded the sections regarding BCC and SCC pathogenesis (sections 2 and 3). For BCC, we added more information on the signaling pathways involved in tumor formation (lines 59-61) and how that impacts tumor progression (lines 61-66). We elaborated on these signaling pathways further in section 2.2, Signaling Pathways in the TME of BCC, in the last paragraph, which discussed how BCC tumor cells directly impact cellular components of the TME, promoting tumor growth (lines 178-186). For SCC, we expanded the introduction similarly. Provided more information about the genetic component of tumor development and signaling in an introductory manner (lines 194-199). In section 3.2, Signaling Pathways in the TME of SCC, we added more information regarding molecular mechanisms influencing tumor progression through the upregulation of specialized membranous extracellular vesicles carrying microRNAs that promote inflammation and tumorigenesis (lines 274-282). For both BCC and SCC, we incorporated schematics summarizing the signaling pathways in the TME, which enhanced those sections and allowed us to condense them slightly.

We appreciate your recommendations for improving the conclusion section with more detail on the therapeutic considerations. The feedback encouraged us to enhance the entire section on Therapeutic Considerations for Skin Cancer to focus more on how research into the TME could provide targets for treatment. We added more papers on the topic and searched clinicaltrials.gov to add a paragraph on novel therapeutic routes (lines 337-363). We took the information from there to write a more detailed conclusion, specifically adding lines 376-382 focused on the therapeutic considerations.

Again, we want to thank you very much for your feedback – we believe the changes we have made upon your recommendation have greatly enhanced the paper.

Round 2

Reviewer 1 Report

Excellent paper, all my concerns were addressed.

Author Response

Great, thank you again for your feedback!

Reviewer 2 Report

The reviewer found the revised version of the manuscript much improved in terms of data presentation and text organization. However, with regard to the need for better insight into the mechanisms involved in the various aspects of the role of the TME, the reviewer suggests adding more recent findings about BCC and SCC pathogenesis. Take into consideration these articles from a midline by PubMed: PMID: 34205977, PMID: 36871014, PMID: 32197405, PMID: 26527779PMID: 26941014. After that, the reviewer retains that manuscript is ready for the publication in Cancers.
